# Modelling Monotonic and Non-Monotonic Attribute Dependencies with Embeddings: A Theoretical Analysis

**Steven Schockaert**                                                          SCHOCKAERTS1@CARDIF.AC.UK

*Cardiff University, UK*

## Abstract

During the last decade, entity embeddings have become ubiquitous in Artificial Intelligence. Such embeddings essentially serve as compact but semantically meaningful representations of the entities of interest. In most approaches, vectors are used for representing the entities themselves, as well as for representing their associated attributes. An important advantage of using attribute embeddings is that (some of the) semantic dependencies between the attributes can thus be captured. However, little is known about what kinds of semantic dependencies can be modelled in this way. The aim of this paper is to shed light on this question, focusing on settings where the embedding of an entity is obtained by pooling the embeddings of its known attributes. Our particular focus is on studying the theoretical limitations of different embedding strategies, rather than their ability to effectively learn attribute dependencies in practice. We first show a number of negative results, revealing that some of the most popular embedding models are not able to capture even basic Horn rules. However, we also find that some embedding strategies are capable, in principle, of modelling both monotonic and non-monotonic attribute dependencies.

## 1. Introduction

Vector space embeddings are currently the dominant representation framework in Natural Language Processing, Computer Vision and Machine Learning. Essentially, these embeddings represent each entity of interest as a dense vector in some fixed-dimensional space. In addition, the attributes that are used to describe these entities are typically also encoded as vectors. For example, in the case of word embeddings, the entities correspond to words and the attributes correspond to the contexts in which these words occur, e.g. in the form of co-occurring words [Mikolov et al., 2013, Pennington et al., 2014], syntactic dependencies [Levy and Goldberg, 2014, Vashishth et al., 2019] or even full sentences [Devlin et al., 2019]. In inductive knowledge graph embedding, the entities from our framework correspond to the previously unseen entities for which we want to learn a representation, with the attributes representing links to known entities [Hamaguchi et al., 2017]. In the case of embedding-based topic models, the entities correspond to documents and the attributes correspond to the associated topics and words [Das et al., 2015, Li et al., 2016, He et al., 2017, Xun et al., 2017, Dieng et al., 2020]. In zero-shot learning, the entities of interest are the category prototypes and the attributes correspond to semantic attributes of the categories [Lampert et al., 2013] or associated natural language terms [Frome et al., 2013].

An important advantage of attribute embeddings is that they can implicitly capture some of the dependencies that hold between the attributes. For instance, the use of embeddings for topic modelling stems from the desire to capture topic correlations [He et al., 2017, Xun et al., 2017]. However, it is currently unclear what kinds of dependencies can be

captured in this way. Before we can address this question, we first need to clarify what it means that an embedding captures some dependency. To this end, consider an embedding model in which $\sigma(\mathbf{a}\cdot\mathbf{e})$ represents the probability that entity $e$ has attribute $a$, where $\sigma$ is the sigmoid function and $\mathbf{a}$ and $\mathbf{e}$ are the embeddings of $a$ and $e$. Among many others, the popular skip-gram word embedding model is of this kind [Mikolov et al., 2013]. Now consider the attributes *parent*, *female* and *mother*, and the associated dependency that *mother* $\equiv$ *parent*$\wedge$*female*. What condition would need to be satisfied for the embedding to capture this dependency? One possibility is to require that $\sigma(\mathbf{mother}\cdot\mathbf{e}) = \sigma(\mathbf{parent}\cdot\mathbf{e})\cdot\sigma(\mathbf{female}\cdot\mathbf{e})$ for each vector $\mathbf{e}$ in the embedding space $\mathbb{R}^n$. However, this condition can clearly not be satisfied; e.g. for $\mathbf{e} = \mathbf{0}$ we obtain the condition $0.5 = 0.25$. Similarly, it is easy to see that the condition $\sigma(\mathbf{mother} \cdot \mathbf{e}) = \min(\sigma(\mathbf{parent} \cdot \mathbf{e}), \sigma(\mathbf{female} \cdot \mathbf{e}))$ only admits trivial solutions. Viewed from this angle, it is clear that popular embedding models are not able to capture even basic logical dependencies. For this reason, several alternative models have been proposed, in which attributes are modelled as linear subspaces [Garg et al., 2019], axis-aligned cones [Vendrov et al., 2016, Özçep et al., 2020], hyperboxes [Vilnis et al., 2018] or polytopes [Gutiérrez-Basulto and Schockaert, 2018], among others.

In this paper, we follow a different direction, analysing whether embedding models can capture logical dependencies in a less demanding sense. In particular, we consider the common setting where the embedding of an entity $e$ has to be learned from its known attributes, and the aim of the resulting embedding is to infer what other attributes $e$ is likely to satisfy. We then say that the rule $a_1 \wedge ... \wedge a_n \to b$ is captured if an entity which is known to have the attributes $a_1, ..., a_n$ would be represented by a vector from which the attribute $b$ would be predicted. Note that different variants of this setting can be considered, which depend on (i) how exactly entity vectors are constructed and (ii) how the resulting vectors are used for predicting the attributes of the entity. The aim of this paper is to analyse, for different variants, whether it is always possible to find an embedding which captures the rule $a_1 \wedge ... \wedge a_n \to b$ iff that rule is entailed by some propositional knowledge base $K$. Note that in practice we typically do not have access to such a knowledge base $K$. However, the question of whether arbitrary propositional knowledge bases can, in principle, be modelled by a given embedding strategy is important, because if this is not the case, then it also means that some (combinations of) dependencies cannot be learned.

In addition to standard propositional entailment, we also look at non-monotonic consequence relations, which is important because attribute dependencies are often defeasible or probabilistic in nature. For instance, in a topic modelling context, we may assume that a document containing the word *safari* is related to the topic *nature*, and a document containing the word *apple* is related to the topic *food*. However, documents containing both of these words are more likely to be related to *technology* instead, given that Safari is the name of Apple's internet browser. We may thus want that the rule *safari* $\to$ *topic:nature* is captured, while the rule *safari* $\wedge$ *apple* $\to$ *topic:nature* is not.

## 2. Problem Setting

We assume that each entity is associated with some attributes from a given set $\mathcal{A}$. In the context of word embeddings, for instance, the set $\mathcal{A}$ could be the set of all contexts. Note that we do not consider knowledge graphs, where entities are described in terms of their

relationship to other entities, rather than attributes. However, many *inductive* knowledge graph embedding models can still be cast as special cases of our framework, with the attributes then representing links to entities with known (or pre-computed) embeddings.

**Embedding and Labelling Functions**  If an entity $e$ is associated with the attributes $a_1, ..., a_n$, we assume that its embedding is given by $\mathbf{e} = Emb(a_1, ..., a_n)$ for some embedding function $Emb : 2^{\mathcal{A}} \to \mathbb{R}^m$. The embedding functions that we consider in this paper will rely on pooling attribute vectors. In particular, we consider embedding functions of the following form: $Emb(a_1, ..., a_n) = \phi(\mathbf{a_1}, ..., \mathbf{a_n})$ for some pooling function $\phi$, where $\mathbf{a} \in \mathbb{R}^m$ represents the embedding of attribute $a \in \mathcal{A}$. For instance, we may have $Emb(a_1, ..., a_n) = \frac{1}{n}(\mathbf{a_1} + ... + \mathbf{a_n})$. Let us furthermore assume that we have a function $Lab : \mathbb{R}^m \to 2^{\mathcal{A}}$ that predicts attributes of entities based on their embeddings. Similar as for $Emb$, we will assume that the labelling function $Lab$ relies on a scoring function that compares the embedding of $e$ with an embedding of the considered attribute. In particular, we consider labelling functions of the following form: $Lab(\mathbf{e}) = \{b \in \mathcal{A} \,|\, \psi(\mathbf{e}, \tilde{\mathbf{b}}) \geq \lambda_b\}$. The embedding $\tilde{\mathbf{b}}$ of the attribute $b$ may in general be different from the attribute embedding $\mathbf{b}$ that is used for $Emb$, similar to how word embedding models learn two types of embeddings for each word. The scalar $\lambda_b$ represents a threshold, which we allow to be attribute-dependent for generality. For instance, we could have $Lab(\mathbf{e}) = \{b \in \mathcal{A} \,|\, \sigma(\mathbf{e} \cdot \mathbf{b}) \geq 0.5\} = \{b \in \mathcal{A} \,|\, \mathbf{e} \cdot \mathbf{b} \geq 0\}$. In the following, we will refer to $(\phi, \psi)$ as an *embedding strategy* and to $(Emb, Lab)$ as an *embedding*. Note that the embedding $(Emb, Lab)$ is determined by the embedding strategy $(\phi, \psi)$, together with the embeddings $\mathbf{a}$ and $\tilde{\mathbf{a}}$ of the attributes in $\mathcal{A}$.

**Consequence Relations**  The function $Lab \circ Emb$ can be viewed as a logical consequence relation. In particular, we say that an embedding $(Emb, Lab)$ captures the rule $a_1 \wedge ... \wedge a_n \to b$ iff $b \in Lab(Emb(a_1, ..., a_n))$, i.e. if an entity that is initially associated with the attributes $a_1, ..., a_n$ would be predicted to have attribute $b$ based on its embedding. Note that under this view, the dependency *mother* $\equiv$ *parent* $\wedge$ *female* can be satisfied for the aforementioned choices of $Emb$ and $Lab$, for instance by choosing $\mathbf{parent} = (-1, 1)$, $\mathbf{female} = (1, 1)$ and $\mathbf{mother} = (0, 1)$. However, now the issue is that a number of unwanted dependencies are also satisfied, such as *female* $\to$ *mother* and *parent* $\to$ *mother*. Therefore, rather than treating rules in isolation, the question we are interested in is the following: given a propositional knowledge base $K$ and a given embedding strategy $(\phi, \psi)$, does there exist a corresponding embedding $(Emb, Lab)$ (or, equivalently, do there exist attribute embeddings) such that for all $b \in \mathcal{A}$ and $\{a_1, ..., a_n\} \subseteq \mathcal{A}$ we have that $b \in Lab(Emb(a_1, ..., a_n))$ iff $K \models a_1 \wedge ... \wedge a_n \to b$ holds, where $\models$ is the standard entailment relation from propositional logic. If this is the case, we say that the embedding strategy $(\phi, \psi)$ can simulate $K$. In Section 4, we will similarly look at whether non-monotonic consequence relations can be simulated with embeddings. Throughout this paper, we will assume that the number of dimensions $m$ can be chosen arbitrarily large, as our focus is on identifying limitations that exist regardless of dimensionality. An overview of our results is shown in Table 1.

## 3. Monotonic Reasoning

In Sections 3.1–3.3, we first discuss embedding strategies which are not capable of capturing certain kinds of propositional knowledge bases. Crucially, these strategies cover many of

| $Emb(a_1, ..., a_n)$ | $Lab(\mathbf{e})$ | **Monotonic** | **Non-mon.** |
|---|---|:---:|:---:|
| $\frac{1}{n}\sum_i \mathbf{a_i}$ | $\{b \mid \mathbf{e}\cdot\tilde{\mathbf{b}} \geq \lambda_b\}$ | ✗ | ✗ |
| $\frac{1}{n}\sum_i \mathbf{a_i}$ | $\{b \mid d(\mathbf{e},\tilde{\mathbf{b}}) \leq \theta_b\}$ | ✗ | ✗ |
| $\frac{\sum_i \mathbf{a_i}}{\|\sum_i \mathbf{a_i}\|}$ | $\{b \mid \mathbf{e}\cdot\tilde{\mathbf{b}} \geq \lambda_b\}$ | ✗ | ✗ |
| $\frac{\sum_i \mathbf{a_i}}{\|\sum_i \mathbf{a_i}\|}$ | $\{b \mid d(\mathbf{e},\tilde{\mathbf{b}}) \leq \theta_b\}$ | ✗ | ✗ |
| $\arg\max_{\mathbf{e}} \sum_i \log \sigma(\mathbf{e}\cdot\mathbf{a_i}) + \kappa\|\mathbf{e}\|^2$ | $\{b \mid \mathbf{e}\cdot\tilde{\mathbf{b}} \geq \lambda_b\}$ | ✗ | ✗ |
| $\arg\max_{\mathbf{e}} \sum_i \log \sigma(\mathbf{e}\cdot\mathbf{a_i}) + \kappa\|\mathbf{e}\|^2$ | $\{b \mid d(\mathbf{e},\tilde{\mathbf{b}}) \leq \theta_b\}$ | ✗ | ✗ |
| $\frac{1}{n}\sum_i \mathbf{a_i}$ | $\{b \mid \text{RELU}(\mathbf{e})\cdot\mathbf{b} \geq 0\}$ | ✓ | ✓ |
| $\mathbf{a_1} \odot ... \odot \mathbf{a_n}$ | $\{b \mid \mathbf{e}\cdot\tilde{\mathbf{b}} \geq 0\}$ | ✓ | ✓ |
| $\mathbf{a_1} \odot ... \odot \mathbf{a_n}$ | $\{b \mid \mathbf{e}\cdot\mathbf{b} \geq 0\}$ | ✗ | ✗ |
| $\max(\mathbf{a_1}, ..., \mathbf{a_n})$ | $\{b \mid \mathbf{b} \preceq \mathbf{e}\}$ | ✓ | ✗ |

Table 1: Overview showing which type of embeddings are able to model monotonic and non-monotonic dependencies.

the most popular embedding models. Section 3.4 then discusses embedding strategies which are capable of modelling arbitrary propositional knowledge bases.

## 3.1 Averaging Based Embeddings

One of the most natural choices for the embedding function $Emb$ consists in averaging the attribute embeddings, i.e.:

$$Emb_{avg}(a_1, ..., a_n) = \frac{1}{n}(\mathbf{a_1} + ... + \mathbf{a_n}) \tag{1}$$

This choice corresponds, among many others, to the common strategy of learning sentence or document vectors by averaging word vectors. It is also closely related to the CBOW model from Mikolov et al. [2013]. As the labelling function, we first consider the common choice to model the probability that entity $e$ has attribute $a$ as $\sigma(\mathbf{e} \cdot \mathbf{a})$. However, since each condition of the form $\sigma(\mathbf{e} \cdot \mathbf{a}) \geq \delta$, with $0 < \delta < 1$, is equivalent to the condition $\mathbf{e} \cdot \mathbf{a} \geq \sigma^{-1}(\delta)$, we use a simple dot product in the formulation:

$$Lab_{dot}(\mathbf{e}) = \{b \in \mathcal{A} \mid \mathbf{e} \cdot \tilde{\mathbf{b}} \geq \lambda_b\} \tag{2}$$

where $\lambda_b \in \mathbb{R}$ is a threshold that may in general be attribute-specific. We will use $\phi_{avg}$ and $\psi_{dot}$ to denote the pooling functions associated with $Emb_{avg}$ and $Lab_{dot}$. For the ease of presentation, throughout the paper, we will similarly use subscripts to link embedding and labelling functions to their corresponding pooling functions, rather than each time introducing these notations explicitly. Note that the embedding strategy $(\phi_{avg}, \psi_{dot})$ also covers models where a linear transformation is applied to the average of the attribute embeddings, as is the case for the *à la carte* method from Khodak et al. [2018].

As the following counterexample shows, not all propositional knowledge bases can be simulated with embeddings of the form $(Emb_{avg}, Lab_{dot})$.

**Counterexample 1.** *Let $K = \{a \wedge b \rightarrow x, c \wedge d \rightarrow x\}$. We show that $K$ cannot be simulated by $(\phi_{avg}, \psi_{dot})$. Indeed, if a suitable embedding existed, among others the following inequalities would have to be satisfied:*

$$\frac{\mathbf{a} + \mathbf{b}}{2} \cdot \tilde{\mathbf{x}} \geq \lambda_x \qquad \frac{\mathbf{c} + \mathbf{d}}{2} \cdot \tilde{\mathbf{x}} \geq \lambda_x \qquad \frac{\mathbf{a} + \mathbf{c}}{2} \cdot \tilde{\mathbf{x}} < \lambda_x \qquad \frac{\mathbf{b} + \mathbf{d}}{2} \cdot \tilde{\mathbf{x}} < \lambda_x$$

*This is not possible, since the first two inequalities imply $(\mathbf{a} + \mathbf{b} + \mathbf{c} + \mathbf{d}) \cdot \tilde{\mathbf{x}} \geq 4\lambda$ whereas the last two imply $(\mathbf{a} + \mathbf{b} + \mathbf{c} + \mathbf{d}) \cdot \tilde{\mathbf{x}} < 4\lambda$.*

Another natural choice for the labelling function is to rely on Euclidean distance:

$$Lab_{dist}(\mathbf{e}) = \{b \in \mathcal{A} \,|\, d(\mathbf{e}, \tilde{\mathbf{b}}) \leq \theta_b\} \tag{3}$$

where $\theta_b \geq 0$. It is easy to verify that the knowledge base from Counterexample 1 can be simulated by choosing $\mathbf{a} = (-1, 0)$, $\mathbf{b} = (1, 0)$, $\mathbf{c} = (0, -1)$, $\mathbf{d} = (0, 1)$ and $\tilde{\mathbf{x}} = (0, 0)$. However, as the following counterexample shows, not all knowledge bases can be simulated using $Emb_{avg}$ and $Lab_{dist}$ either.

**Counterexample 2.** *Let $K = \{a \wedge b \rightarrow x, c \wedge d \rightarrow x, a \wedge c \rightarrow y, b \wedge d \rightarrow y\}$. We show that $K$ cannot be simulated by $(\phi_{avg}, \psi_{dist})$. Indeed, suppose that a suitable embedding $(Emb_{avg}, Lab_{dist})$ existed. From the rules with $x$ in the head, it follows that $d^2\left(\frac{\mathbf{a}+\mathbf{b}}{2}, \tilde{\mathbf{x}}\right) + d^2\left(\frac{\mathbf{c}+\mathbf{d}}{2}, \tilde{\mathbf{x}}\right) < d^2\left(\frac{\mathbf{a}+\mathbf{c}}{2}, \tilde{\mathbf{x}}\right) + d^2\left(\frac{\mathbf{b}+\mathbf{d}}{2}, \tilde{\mathbf{x}}\right)$. This is equivalent with:*

$$\|\mathbf{a}\|^2 + \|\mathbf{b}\|^2 + \|\mathbf{c}\|^2 + \|\mathbf{d}\|^2 + 8\|\mathbf{x}\|^2 + 2(\mathbf{a} \cdot \mathbf{b} + \mathbf{c} \cdot \mathbf{d}) - 4(\mathbf{a} + \mathbf{b} + \mathbf{c} + \mathbf{d}) \cdot \tilde{\mathbf{x}}$$
$$< \|\mathbf{a}\|^2 + \|\mathbf{c}\|^2 + \|\mathbf{b}\|^2 + \|\mathbf{d}\|^2 + 8\|\mathbf{x}\|^2 + 2(\mathbf{a} \cdot \mathbf{c} + \mathbf{b} \cdot \mathbf{d}) - 4(\mathbf{a} + \mathbf{b} + \mathbf{c} + \mathbf{d}) \cdot \tilde{\mathbf{x}}$$

*which simplifies to $\mathbf{a} \cdot \mathbf{b} + \mathbf{c} \cdot \mathbf{d} < \mathbf{a} \cdot \mathbf{c} + \mathbf{b} \cdot \mathbf{d}$. In the same way, using the rules with $y$ in the head, we find $\mathbf{a} \cdot \mathbf{b} + \mathbf{c} \cdot \mathbf{d} > \mathbf{a} \cdot \mathbf{c} + \mathbf{b} \cdot \mathbf{d}$, which is a contradiction.*

To illustrate the relevance of these results, let the attributes $a_1, ..., a_n$ correspond to the words that are observed in a document $d$, and suppose that other attributes, which are not observed, correspond to document categories. We may want the embedding model to capture rules such as *tennis $\wedge$ player $\wedge$ won $\rightarrow$ cat:sports*, meaning that if the words *tennis*, *player* and *won* appear in a document, then it should belong to the category *sports*. Our results show that, in general, such dependencies cannot be captured when $Emb_{avg}$ is used in combination with $Lab_{dot}$ or $Lab_{dist}$. For instance, this means that there are theoretical limitations to the kinds of categories that may be predicted from document embeddings when using the *à la carte* method from [Khodak et al., 2018] together with a linear classifier.

### 3.2 Embeddings as Normalised Averages

Let us now consider the following embedding function, which represents entities using normalised vectors:

$$Emb_{norm}(a_1, ..., a_n) = \frac{\mathbf{a_1} + ... + \mathbf{a_n}}{\|\mathbf{a_1} + ... + \mathbf{a_n}\|} \tag{4}$$

provided $\|\mathbf{a_1} + ... + \mathbf{a_n}\| > 0$, and $Emb_{norm}(a_1, ..., a_n) = \mathbf{0}$ otherwise. Entity vectors can then be understood as maximum likelihood estimates, if we view attributes as von Mises-Fisher distributions, which is a common choice when modelling text [Banerjee et al., 2005,

Batmanghelich et al., 2016, Meng et al., 2019]. In particular, if $p(\mathbf{e}|a)$ is a von Mises-Fisher distribution with mean $\frac{\mathbf{a}}{\|\mathbf{a}\|}$ and concentration parameter $\|\mathbf{a}\|$, for each $a \in \mathcal{A}$, then we have:

$$Emb_{norm}(a_1, ..., a_n) = \arg\max_{\mathbf{e}} \prod_{i=1}^{n} p(\mathbf{e}|a_i) \quad \text{s.t. } \|\mathbf{e}\| = 1$$

$$= \arg\max_{\mathbf{e}} \sum_{i=1}^{n} \mathbf{e} \cdot \mathbf{a_i} \quad \text{s.t. } \|\mathbf{e}\| = 1$$

The question of whether propositional knowledge bases can be simulated with $Emb_{norm}$ is thus relevant for understanding the limitations of von Mises-Fisher based topic models and document representations [Meng et al., 2019, Batmanghelich et al., 2016]. The following counterexample shows that not all knowledge bases can be simulated with $Emb_{norm}$. While the use of normalised averages is intuitively similar to the averages from Section 3.1, the counterexample in this case is more involved.

**Counterexample 3.** *Let $K = \{a \wedge b \to x, c \wedge d \to x, a \wedge c \to y, b \wedge d \to y, a \wedge d \to y, b \wedge c \to y\}$. We show that $K$ cannot be simulated by $(\phi_{norm}, \psi_{dot})$. Indeed, suppose that a suitable embedding $(Emb_{norm}, Lab_{dot})$ existed. Then we must have $\lambda_x > 0$ and $\lambda_y > 0$, which follows immediately from the fact that $Emb_{norm}(\emptyset) = \mathbf{0}$ while $K \not\models \top \to x$ and $K \not\models \top \to y$. For the ease of presentation, let us introduce the following abbreviations: $\mathbf{y}_{ab} = \frac{\mathbf{a}+\mathbf{b}}{\|\mathbf{a}+\mathbf{b}\|}$ and similar for $\mathbf{y}_{ac}, \mathbf{y}_{ad}, \mathbf{y}_{bc}, \mathbf{y}_{bd}, \mathbf{y}_{cd}$. Consider the hyperplane $H$ defined by $H = \{\mathbf{e} \,|\, \mathbf{e} \cdot (\lambda_x \tilde{\mathbf{y}} - \lambda_y \tilde{\mathbf{x}}) = 0\}$. If $\mathbf{e}$ belongs to the positive half-space $H^+ = \{\mathbf{e} \,|\, \mathbf{e} \cdot (\lambda_x \tilde{\mathbf{y}} - \lambda_y \tilde{\mathbf{x}}) \geq 0\}$, we have:*

$$\mathbf{e} \cdot \tilde{\mathbf{y}} \geq \frac{\lambda_y}{\lambda_x} \mathbf{e} \cdot \tilde{\mathbf{x}}$$

*This implies that either $\mathbf{e} \cdot \tilde{\mathbf{x}} < \lambda_x$ or $\mathbf{e} \cdot \tilde{\mathbf{y}} \geq \lambda_y$. We thus find in particular that $\mathbf{y}_{ab}$ and $\mathbf{y}_{cd}$ do not belong to this positive half-space $H^+$. In the same way, we find that $\mathbf{y}_{ac}, \mathbf{y}_{bd}, \mathbf{y}_{ad}$ and $\mathbf{y}_{bc}$ do not belong to the negative half-space $H^- = \{\mathbf{e} \,|\, \mathbf{e} \cdot (\lambda_x \tilde{\mathbf{y}} - \lambda_y \tilde{\mathbf{x}}) \leq 0\}$. Note that when $\mathbf{e}_1, \mathbf{e}_2 \in H^+$, we also have $\mathbf{e}_1 + \mathbf{e}_2 \in H^+$ and $\frac{\mathbf{e}_1+\mathbf{e}_2}{\|\mathbf{e}_1+\mathbf{e}_2\|} \in H^+$, and similar for $H^-$. Since $\mathbf{y}_{ab}, \mathbf{y}_{cd} \in H^-$, at least one of $\mathbf{a}, \mathbf{b}$ must thus belong to $H^-$ and at least one of $\mathbf{c}, \mathbf{d}$ must belong to $H^-$. Assume for instance that $\mathbf{a} \in H^-$ and $\mathbf{c} \in H^-$; the other cases follow by symmetry. From $\mathbf{a} \in H^-$ and $\mathbf{c} \in H^-$, we find that $\mathbf{y}_{ac} \in H^-$, which is a contradiction.*

While we used $Lab_{dot}$ in the above counterexample, the result also holds for $Lab_{dist}$. Indeed, since $\|\mathbf{e}\| = 1$, we have $d(\mathbf{e}, \tilde{\mathbf{b}}) \leq \theta_b$ iff $d^2(\mathbf{e}, \tilde{\mathbf{b}}) \leq \theta_b^2$ iff $\mathbf{e} \cdot \tilde{\mathbf{b}} \geq \frac{1}{2}(1 + \|\tilde{\mathbf{b}}\|^2 - \theta_b^2)$. We thus have that $K$ can be simulated by $(\phi_{norm}, \psi_{dot})$ iff it can be simulated by $(\phi_{norm}, \psi_{dist})$.

### 3.3 Sigmoid Based Embeddings

We now turn to the common choice of modelling attribute probabilities using the sigmoid function, i.e. let us assume that $\sigma(\mathbf{e} \cdot \mathbf{a})$ represents the probability that entity $e$ has attribute $a$. A natural choice for inferring the embedding of $e$ is then to maximise the likelihood of the observed attributes $a_1, ..., a_n$:

$$Emb_{sig}(a_1, ..., a_n) = \arg\max_{\mathbf{e}} \sum_{i=1}^{n} \log \sigma(\mathbf{e} \cdot \mathbf{a_i}) + \kappa \|\mathbf{e}\|^2 \tag{5}$$

where $\kappa > 0$ is a constant. This embedding strategy closely corresponds to the skip-gram model [Mikolov et al., 2013], with two differences. First, the standard skip-gram model does not include the regularisation term $\kappa\|\mathbf{e}\|^2$. Here, we need to add this term, which amounts to imposing a Gaussian prior, to ensure that $Emb_{sig}$ is well-defined[1]. Second, the skip-gram model also includes negative samples. However, since $1 - \sigma(\mathbf{e} \cdot \mathbf{a}) = \sigma(-\mathbf{e} \cdot \mathbf{a})$, negative samples can be considered as a special case where some of the attributes $a_i$ capture the fact that another attribute $b_i$ is not present, by constraining the embedding such that $\mathbf{a_i} = -\mathbf{b_i}$. The limitations of the embedding function $Emb_{sig}$ thus still apply to settings where negative samples are used. The following counterexample shows that arbitrary propositional knowledge bases cannot be modelled with $Emb_{sig}$ and $Lab_{dot}$.

**Counterexample 4.** *The main idea is to follow the same strategy as in Counterexample 3, which is possible thanks to the fact that $Emb_{sig}(a,b)$ is a conical combination of $Emb_{sig}(a)$ and $Emb_{sig}(b)$, provided $\cos(\mathbf{a}, \mathbf{b}) > -1$. Some care is needed to ensure that the latter condition is satisfied for various pairs of attributes.*

*Let $K = \{a \wedge b \to x, c \wedge d \to x, a \wedge c \to y, b \wedge d \to y, a \wedge d \to y, b \wedge c \to y, a \to z_{ab}, b \to z_{ab}, a \to z_{ac}, c \to z_{ac}, a \to z_{ad}, d \to z_{ad}, b \to z_{bc}, c \to z_{bc}, b \to z_{bd}, d \to z_{bd}, c \to z_{cd}, d \to z_{cd}\}$. We show that $K$ cannot be simulated by $(\phi_{sig}, \psi_{dot})$. Suppose that a suitable embedding $(Emb_{sig}, Lab_{dot})$ existed. Note that $\mathbf{y_a} = \alpha_a \mathbf{a}$ and $\mathbf{y_b} = \alpha_b \mathbf{b}$ for some $\alpha_a, \alpha_b > 0$. Since we have $\lambda_{z_{ab}} > 0$ (which follows in the same way as $\lambda_x > 0$), it must be the case that $\cos(\mathbf{y_a}, \tilde{\mathbf{z}}_{\mathbf{ab}}) > 0$ and $\cos(\mathbf{y_b}, \tilde{\mathbf{z}}_{\mathbf{ab}}) > 0$, which means $\cos(\mathbf{a}, \tilde{\mathbf{z}}_{\mathbf{ab}}) > 0$ and $\cos(\mathbf{b}, \tilde{\mathbf{z}}_{\mathbf{ab}}) > 0$, which implies $\cos(\mathbf{a}, \mathbf{b}) > -1$. This, in turn, implies that $\mathbf{y_{ab}}$ is a conical combination of $\mathbf{y_a}$ and $\mathbf{y_b}$. We similarly have that $\mathbf{y_{ac}}$, $\mathbf{y_{ad}}$, $\mathbf{y_{bc}}$, $\mathbf{y_{bd}}$ and $\mathbf{y_{cd}}$ are conical combinations of the corresponding attribute vectors. Consider again the hyperplane $H$ defined by $H = \{\mathbf{e} \,|\, \mathbf{e} \cdot (\lambda_x \tilde{\mathbf{y}} - \lambda_y \tilde{\mathbf{x}}) = 0\}$ from Counterexample 3, and the positive and negative half-spaces $H^+$ and $H^-$. When $\mathbf{e}_1, \mathbf{e}_2 \in H^+$, we also have $\mu_1 \mathbf{e}_1 + \mu_2 \mathbf{e}_2 \in H^+$ for $\mu_1, \mu_2 \geq 0$, i.e. if $\mathbf{e}_1$ and $\mathbf{e}_2$ are in $H^+$ then the same is true for any conical combination of $e_1$ and $e_2$, and similar for $H^-$. We thus obtain a contradiction in the same way as in Counterexample 3.*

In the appendix, we provide a similar counterexample for the strategy $(\phi_{sig}, \psi_{dist})$. Note that Counterexample 4 only relies on the fact that $Emb_{sig}(a_1, ..., a_n)$ is a conical combination of $\mathbf{a_1}, ..., \mathbf{a_n}$. The same counterexample can thus be used for other embeddings strategies that rely on a weighted average of attribute vectors with non-negative weights.

### 3.4 Modelling Monotonic Dependencies with Embeddings

Thus far, we have found that standard embedding strategies are not capable of simulating even basic sets of Horn rules. It turns out, however, that this limitation can be solved by adding a non-linearity to the labelling function:

$$Lab_{relu}(\mathbf{e}) = \{b \in \mathcal{A} \,|\, \text{ReLU}(\mathbf{e}) \cdot \mathbf{b} \geq 0\} \tag{6}$$

where the ReLU function is applied component-wise. Note that in the definition of $Lab_{relu}$ we fixed the threshold at 0 and we fixed $\mathbf{a} = \tilde{\mathbf{a}}$ for all $a \in \mathcal{A}$, to highlight the fact that this restricted definition is already sufficient for modelling propositional dependencies.

---

1. In particular, this ensures that the maximum is attained for a vector with finite coordinates. This vector may not be unique, however, in which case we assume that an arbitrary maximising vector $\mathbf{e}$ is chosen.

**Proposition 1.** *For any propositional knowledge base $K$ over $\mathcal{A}$, there exist embeddings of the attributes in $\mathcal{A}$ such that $b \in Lab_{relu}(Emb_{avg}(a_1, ..., a_n))$ iff $K \models a_1 \wedge ... \wedge a_n \rightarrow b$.*

*Proof.* Let $mod(K) = \{\omega_1, ..., \omega_l\}$ be the set of models of $K$. We define the embedding $\mathbf{a} = (x_1^a, ..., x_{l+1}^a)$ of the attribute $a$ as follows:

$$
x_i^a = \begin{cases} 1 & \text{if } i \leq l \text{ and } \omega_i \models a \\ -\delta & \text{if } i \leq l \text{ and } \omega_i \not\models a \\ 1 & \text{if } i = l+1 \end{cases}
$$

where $\delta$ is a constant which is chosen such that $\delta > 2^{|\mathcal{A}|}$. Let $\mathbf{e} = (y_1, ..., y_{l+1}) = Emb_{avg}(a_1, ..., a_n)$. Let $n_i$ be the number of atoms from $\{a_1, ..., a_n\}$ that are satisfied in $\omega_i$. Then we have $y_{l+1} = 1$ and for $i \leq l$ we have

$$
y_i = \frac{1}{n}\left(n_i - (n - n_i)\delta\right)
$$

In particular, we have $y_i = 1$ if $\omega_i \models \{a_1, ..., a_n\}$ and $y_i < 0$ otherwise (since $\delta > |\mathcal{A}| \geq n_i$). We thus have:

$$
\text{RELU}(y_i) = \begin{cases} 1 & \text{if } \omega_i \models \{a_1, ..., a_n\} \\ 0 & \text{otherwise} \end{cases}
$$

For $b \in \mathcal{A}$, we find

$$
\text{RELU}(\mathbf{e}) \cdot \mathbf{b} = 1 + |\{\omega \mid \omega \models K \cup \{a_1, ..., a_n, b\}\}| - \delta|\{\omega \mid \omega \models K \cup \{a_1, ..., a_n, \neg b\}\}|
$$

In particular, we have $\text{RELU}(\mathbf{e}) \cdot \mathbf{b} \geq 0$ iff $|\{\omega \mid \omega \models K \cup \{a_1, ..., a_n, \neg b\}\}| = 0$, which is equivalent to $K \models a_1 \wedge ... \wedge a_n \rightarrow b$. $\square$

Note that the proof can be straightforwardly adapted to other types of non-linearities. For instance, with sigmoid instead of ReLU, by choosing $\delta$ sufficiently large, we can ensure that $\sigma(y_i)$ is arbitrarily close to 0 when $\omega_i \not\models \{a_1, ..., a_n\}$, and rely on the same argument.

Next we consider the following embedding function:

$$
Emb_{had}(a_1, ..., a_n) = \mathbf{a_1} \odot ... \odot \mathbf{a_n} \tag{7}
$$

where we write $\odot$ for the Hadamard product (i.e. the component-wise product of vectors). In the appendix, we show the following result, using a construction that is very similar to the one from the proof of Proposition 1.

**Proposition 2.** *For any propositional knowledge base $K$ over $\mathcal{A}$, there exist embeddings of the attributes in $\mathcal{A}$ such that $b \in Lab_{dot}(Emb_{had}(a_1, ..., a_n))$ iff $K \models a_1 \wedge ... \wedge a_n \rightarrow b$.*

Note that in contrast to the setting from Proposition 1, here we allow $\mathbf{a} \neq \tilde{\mathbf{a}}$. It is easy to see that this additional freedom is necessary, since otherwise any embedding modelling a rule of the form $a \rightarrow b$ would also model the reversed rule $b \rightarrow a$. Finally, we consider the embedding strategy that is used in the order embeddings from Vendrov et al. [2016]:

$$
Emb_{ord}(a_1, ..., a_n) = \max(\mathbf{a_1}, ..., \mathbf{a_n}) \qquad Lab_{ord}(\mathbf{e}) = \{b \in \mathcal{A} \mid \mathbf{b} \preceq \mathbf{e}\}
$$

where we write max for the component-wise maximum of the vectors and $\preceq$ is the product order, i.e. $(x_1, ..., x_n) \preceq (y_1, ..., y_n)$ iff $x_i \leq y_i$ for all $i \in \{1, ..., n\}$. This embedding model was proposed to improve how hierarchical relations can be encoded. However, as the next result shows, it also allows us to simulate other kinds of propositional dependencies.

**Proposition 3.** *For any propositional knowledge base $K$ over $\mathcal{A}$, there exist embeddings of the attributes in $\mathcal{A}$ such that $b \in Lab_{ord}(Emb_{ord}(a_1, ..., a_n))$ iff $K \models a_1 \wedge ... \wedge a_n \to b$.*

*Proof.* Let $mod(K) = \{\omega_1, ..., \omega_l\}$ be the set of models of $K$. If $K$ is inconsistent, we can simply choose $\mathbf{a} = \mathbf{0}$ for every $a \in \mathcal{A}$, with $\mathbf{0}$ a vector of zeroes of an arbitrary dimension. Now suppose $|mod(K)| > 0$. We define the embedding $\mathbf{a} = (x_1^a, ..., x_l^a)$ of the attribute $a$ as follows:

$$x_i^a = \begin{cases} 0 & \text{if } \omega_i \models a \\ 1 & \text{otherwise} \end{cases}$$

Let $\mathbf{e} = (y_1, ..., y_l) = Emb_{ord}(a_1, ..., a_n)$. Then we have $y_i = 0$ if $\omega_i \models \{a_1, ..., a_n\}$ and $y_i = 1$ otherwise. We thus have $\mathbf{b} \preceq \mathbf{e}$ iff $\forall i.(x_i^b = 1) \Rightarrow (y_i = 1)$ iff $\forall i.(\omega_i \not\models b) \Rightarrow (\omega \not\models \{a_1, ..., a_n\})$ iff $K \models a_1 \wedge ... \wedge a_n \to b$. $\square$

Order embeddings essentially represent each attribute as an axis-aligned cone (where the vector components are viewed as lower bounds). Other region based embedding models can be used to model monotonic dependencies in a similar way, including e.g. box embeddings [Vilnis et al., 2018] and hyperbolic entailment cones [Ganea et al., 2018].

## 4. Non-Monotonic Reasoning

We now consider a standard ranking-based semantics of default rules [Lehmann and Magidor, 1992]. Let $\Theta$ be a stratified knowledge base, i.e. a ranked list of formulas $(\alpha_1, ..., \alpha_k)$. Then we say that $\Theta \models \alpha \vartriangleright \beta$ iff there is some $i \in \{0, ..., k\}$ such that $\alpha_1 \wedge ... \wedge \alpha_i \wedge \alpha \models \beta$ and $\alpha_1 \wedge ... \wedge \alpha_i \wedge \alpha \not\models \neg\beta$. The formula $\alpha \vartriangleright \beta$ intuitively means that "if $\alpha$ holds then typically also $\beta$ holds". This semantics of default rules can equivalently be characterised in terms of the *maximum a posteriori* (MAP) consequences of a probabilistic model, i.e. $\alpha \vartriangleright \beta$ is inferred iff $\beta$ is true in the most probable models of $\alpha$ [Kuzelka et al., 2016].

**Example 1.** *Consider the following stratified knowledge base:*

$$\Theta = (\neg cat{:}technology \vee \neg cat{:}food, apple \wedge safari \to cat{:}technology, apple \to cat{:}food)$$

*It can be verified that $\Theta \models apple \vartriangleright cat{:}food$ and $\Theta \models apple \wedge safari \vartriangleright cat{:}technology$, while $\Theta \not\models apple \wedge safari \vartriangleright cat{:}food$.*

We now analyse whether an embedding can be found such that $b \in Lab(Emb(a_1, ..., a_n))$ iff $\Theta \models a_1 \wedge ... \wedge a_n \vartriangleright b$. Clearly, embedding strategies which cannot be used to simulate monotonic reasoning cannot be used to simulate this form of non-monotonic reasoning either. This is because we can choose $\Theta_1 = (\alpha_1)$, where $\alpha_1$ is the conjunction of all formulas in a propositional knowledge base $K$. However, we find that the strategy $(\phi_{avg}, \psi_{relu})$ can be used to model non-monotonic attribute dependencies.

**Proposition 4.** *For any stratified knowledge base $\Theta$ over $\mathcal{A}$, there exist embeddings of the attributes in $\mathcal{A}$ such that $b \in Lab_{relu}(Emb_{avg}(a_1, ..., a_n))$ iff $\Theta \models a_1 \wedge ... \wedge a_n \rhd b$.*

*Proof.* Let $\Theta = (\alpha_1, ..., \alpha_m)$ and let $\omega_1, ..., \omega_l$ be an enumeration of all interpretations over $\mathcal{A}$. To define the embeddings $\mathbf{a} = (x_1^a, ..., x_l^a)$, we use the mapping $\mu$ defined by $\mu(\omega) = \max\{i \,|\, \omega \models \alpha_1 \wedge ... \wedge \alpha_i\}$, where we assume $\mu(\omega) = 0$ if $\omega \not\models \alpha_1$. We define:

$$x_i^a = \begin{cases} \delta^{2\mu(\omega_i)} & \text{if } \omega_i \models a \\ -\delta^{(1+2\mu(\omega_i))} & \text{otherwise} \end{cases}$$

where $\delta$ is chosen such that $\delta > 2^{|\mathcal{A}|}$. Let $\mathbf{e} = (y_1, ..., y_l) = Emb_{avg}(a_1, ..., a_n)$. Similar as in the proof of Proposition 1 we then find $y_i < 0$ iff $\omega_i \not\models \{a_1, ..., a_n\}$, i.e.:

$$\text{RELU}(y_i) = \begin{cases} \delta^{2\mu(\omega_i)} & \text{if } \omega_i \models \{a_1, ..., a_n\} \\ 0 & \text{otherwise} \end{cases}$$

For $b \in \mathcal{A}$, we find that $\text{RELU}(Emb_{avg}(a_1, ..., a_n)) \cdot \mathbf{b}$ is given by

$$\sum_{\omega_i \models \{a_1,...,a_n,b\}} \delta^{4\mu(\omega_i)} - \sum_{\omega_i \models \{a_1,...,a_n,\neg b\}} \delta^{(1+4\mu(\omega_i))}$$

We thus have $\text{RELU}(Emb_{avg}(a_1, ..., a_n)) \cdot \mathbf{b} \geq 0$ iff

$$\sum_{\omega_i \models \{a_1,...,a_n,b\}} \delta^{4\mu(\omega_i)} \geq \sum_{\omega_i \models \{a_1,...,a_n,\neg b\}} \delta^{(1+4\mu(\omega_i))} \tag{8}$$

Let $m^+ = \max\{\mu(\omega_i)|\omega_i \models \{a_1, ..., a_n, b\}\}$ and $m^- = \max\{\mu(\omega_i)|\omega_i \models \{a_1, ..., a_b, \neg b\}\}$, where we define $m^- = -1$ if $\{a_1, ..., a_n, \neg b\}$ is inconsistent (i.e. if $b = \neg a_i$ for some $i$). Then we have that $\Theta \models a_1 \wedge ... \wedge a_n \rhd b$ is equivalent to $m^+ > m^-$. If $m^+ > m^-$, we find

$$\sum_{\omega_i \models \{a_1,...,a_n,b\}} \delta^{4\mu(\omega_i)} \geq \delta^{4m^+} = \delta^3 \cdot \delta^{1+4(m^+-1)} > \delta \cdot \delta^{1+4(m^+-1)} > 2^{|At|} \cdot \delta^{1+4(m^+-1)}$$

$$\geq 2^{|At|} \cdot \delta^{1+4(m^-)} \geq \sum_{\omega_i \models \{a_1,...,a_n,\neg b\}} \delta^{(1+4\mu(\omega_i))}$$

Conversely, if $m^+ \leq m^-$ we find

$$\sum_{\omega_i \models \{a_1,...,a_n,b\}} \delta^{4\mu(\omega_i)} \leq \sum_{\omega_i \models \{a_1,...,a_n,b\}} \delta^{4m^+} \leq 2^{|At|} \cdot \delta^{4m^+} < \delta \cdot \delta^{4m^+} = \delta^{1+4m^+} \leq \delta^{1+4m^-}$$

$$\leq \sum_{\omega_i \models \{a_1,...,a_n,\neg b\}} \delta^{(1+4\mu(\omega_i))}$$

where the last step relies on the fact that $m^+ \leq m^-$ implies $m^- \geq 0$ and thus there must be some $\omega_i$ such that $\omega_i \models \{a_1, ..., a_n, b\}$. We thus have that $\Theta \models a_1 \wedge ... \wedge a_n \rhd b$ is equivalent to $m^+ > m^-$, which is equivalent to (9) and $\text{RELU}(Emb_{avg}(a_1, ..., a_n)) \cdot \mathbf{b} \geq 0$. $\qquad \square$

In the appendix we show the following result, using a similar construction.

**Proposition 5.** *For any stratified knowledge base $\Theta$ over $\mathcal{A}$, there exist embeddings of the attributes in $\mathcal{A}$ such that $b \in Lab_{dot}(Emb_{had}(a_1,...,a_n))$ iff $\Theta \models a_1 \wedge ... \wedge a_n \rhd b$.*

Finally, we show that the combination $Emb_{ord}$ and $Lab_{ord}$ cannot be used to model non-monotonic dependencies. A similar limitation arises for all approaches which learn embeddings by taking the intersection of region-based attribute representations.

**Counterexample 5.** *Let $\Theta = (a \wedge b \rightarrow \bot, a \rightarrow x)$. To simulate $\Theta$ with an embedding of the form $(Emb_{ord}, Lab_{ord})$, we need $\mathbf{x} \preceq \mathbf{a}$ and $\mathbf{x} \npreceq \max(\mathbf{a}, \mathbf{b})$. However, this is impossible since $\mathbf{a} \preceq \max(\mathbf{a}, \mathbf{b})$.*

## 5. Concluding Remarks

It remains poorly understood how we can design embedding models to encourage different kinds of dependencies to be captured (with the problem of embedding hierarchies being a notable exception [Vendrov et al., 2016, Nickel and Kiela, 2017, Ganea et al., 2018]). The analysis presented in this paper provides a step towards such an understanding. The ability of the ReLU-based labelling function to model monotonic and non-monotonic attribute dependencies seems of particular interest, given how close it stays to standard embedding models. While we have focused on propositional dependencies, our results also have implications for knowledge graph embedding. For instance, bilinear models can be viewed as instances of the sigmoid based embedding strategy, where attributes represent links to other entities. The attribute vectors then depend on the embeddings of other entities, which are iteratively updated. This makes it possible to capture certain types of relational dependencies. Developing a better understanding of which types of relational dependencies can be modelled in this way is an important avenue for future work. However, the results presented in this paper already show that such models are not able to capture arbitrary dependencies.

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

## Appendix A. Counterexample for $Emb_{sig}$ with $Lab_{dist}$

Let $K$ be defined as in Counterexample 4, and let us again use the abbreviations of the form $\mathbf{y_{ab}}$ and $\mathbf{y_a}$. We show that there must always exist a hyperplane $H$ that separates $\mathbf{y_{ab}}$ and $\mathbf{y_{cd}}$, on the one hand, from $\mathbf{y_{ac}}$, $\mathbf{y_{ad}}$, $\mathbf{y_{bc}}$ and $\mathbf{y_{bd}}$, on the other hand. The fact that $K$ cannot be modelled using $Emb_{sig}(a)$ and $Lab_{dist}$ then follows in the same way as in Counterexample 4.

Let us write $S_x$ for the hypersphere around $\tilde{\mathbf{x}}$ of radius $\theta_x$, i.e. $S_x = \{\mathbf{e} \,|\, d(\mathbf{e}, \tilde{\mathbf{x}}) \leq \theta_x\}$, and let $S_y$ similarly be the hypersphere around $\tilde{\mathbf{y}}$ of radius $\theta_y$. If $d(\tilde{x}, \tilde{u}) \geq \theta_x + \theta_y$, then we can simply define $H$ as any hyperplane that separates $S_x$ and $S_y$. If $d(\tilde{x}, \tilde{u}) < \theta_x + \theta_y$, then we choose $H$ as the unique hyperplane that contains the intersection of the boundaries of $S_x$ and $S_y$, noting that this boundary cannot be empty, since $\mathbf{y}_{ab}, \mathbf{y}_{cd} \in S_x \setminus S_y$ and $\mathbf{y}_{ac}, \mathbf{y}_{ad}, \mathbf{y}_{bc}, \mathbf{y}_{bd} \in S_y \setminus S_x$, meaning that we cannot have $S_x \subseteq S_y$ or $S_y \subseteq S_x$. Moreover, since $\mathbf{y}_{ab}, \mathbf{y}_{cd} \in S_x \setminus S_y$ and $\mathbf{y}_{ac}, \mathbf{y}_{ad}, \mathbf{y}_{bc}, \mathbf{y}_{bd} \in S_y \setminus S_x$, in both cases we clearly have that $H$ separates $\mathbf{y}_{ab}, \mathbf{y}_{cd}$ from $\mathbf{y}_{ac}, \mathbf{y}_{ad}, \mathbf{y}_{bc}, \mathbf{y}_{bd}$.

## Appendix B. Proof of Proposition 2

Let $mod(K) = \{\omega_1, ..., \omega_l\}$ be the set of models of $K$. We define the embeddings $\mathbf{a} = (x_1^a, ..., x_{l+1}^a)$ and $\tilde{\mathbf{a}} = (\tilde{x}_1^a, ..., \tilde{x}_{l+1}^a)$ of the atom $a$ as follows:

$$x_i^a = \begin{cases} 1 & \text{if } i \leq l \text{ and } \omega_i \models a \\ 0 & \text{if } i \leq l \text{ and } \omega_i \not\models a \\ 1 & \text{if } i = l+1 \end{cases} \qquad \tilde{x}_i^a = \begin{cases} 1 & \text{if } i \leq l \text{ and } \omega_i \models a \\ -\delta & \text{if } i \leq l \text{ and } \omega_i \not\models a \\ 1 & \text{if } i = l+1 \end{cases}$$

where $\delta$ is a constant satisfying $\delta > 2^{|\mathcal{A}|}$. Let us write $Emb_{had}(a_1, ..., a_n) = \mathbf{y} = (y_1, ..., y_{l+1})$. Then we clearly have:

$$y_i = \begin{cases} 1 & \text{if } \omega_i \models \{a_1, ..., a_n\} \\ 0 & \text{otherwise} \end{cases}$$

For $b \in \mathcal{A}$, we find

$$\mathbf{y} \cdot \tilde{\mathbf{b}} = 1 + |\{\omega \,|\, \omega \models K \cup \{a_1, ..., a_n, b\}\}| - \delta|\{\omega \,|\, \omega \models K \cup \{a_1, ..., a_n, \neg b\}\}|$$

Note that we have $\mathbf{y} \cdot \tilde{\mathbf{b}} \geq 0$ iff $|\{\omega \,|\, \omega \models K \cup \{a_1, ..., a_n, \neg b\}\}| = 0$, since we assumed $\delta > 2^{|\mathcal{A}|}$. We have $|\{\omega \,|\, \omega \models K \cup \{a_1, ..., a_n, \neg b\}\}| = 0$ iff $K \cup \{a_1, ..., a_n\} \models b$ iff $K \models a_1 \wedge ... \wedge a_n \to b$. In particular, we have:

$$(Emb_{had}(a_1, ..., a_n) \cdot \tilde{\mathbf{b}} \geq 0) \quad \Leftrightarrow \quad (K \models a_1 \wedge ... \wedge a_n \to b)$$

## Appendix C. Proof of Proposition 5

Let $\Omega = (\alpha_1, .., \alpha_m)$ and let $\omega_1, ..., \omega_l$ be an enumeration of all interpretations over $\mathcal{A}$. We define the embedding $\mathbf{a} = (x_1^a, ..., x_l^a)$ as follows:

$$x_i^a = \begin{cases} 1 & \text{if } \omega_i \models a \\ 0 & \text{otherwise} \end{cases}$$

To define the embedding $\tilde{\mathbf{a}} = (\tilde{x}_1^a, ..., \tilde{x}_l^a)$, we use the mapping $\mu$ defined by $\mu(\omega) = \max\{i \,|\, \omega \models \alpha_1 \wedge ... \wedge \alpha_i\}$, where we assume $\mu(\omega) = 0$ if $\omega \not\models \alpha_1$:

$$\tilde{x}_i^a = \begin{cases} \delta^{2\mu(\omega_i)} & \text{if } \omega_i \models a \\ -\delta^{(1+2\mu(\omega_i))} & \text{otherwise} \end{cases}$$

where $\delta$ is a constant which is chosen such that $\delta > 2^{|\mathcal{A}|}$. Let us write $Emb_{had}(a_1, ..., a_n) = \mathbf{y} = (y_1, ..., y_l)$. Note that we have:

$$y_i = \begin{cases} 1 & \text{if } \omega_i \models \{a_1, ..., a_n\} \\ 0 & \text{otherwise} \end{cases}$$

For $b \in \mathcal{A}$, we find

$$\mathbf{y} \cdot \tilde{\mathbf{b}} = \sum_{\omega_i \models \{a_1, ..., a_n, b\}} \delta^{2\mu(\omega_i)} - \sum_{\omega_i \models \{a_1, ..., a_n, \neg b\}} \delta^{(1+2\mu(\omega_i))}$$

We thus have $\mathbf{y} \cdot \tilde{\mathbf{b}} \geq 0$, i.e. $b \in Lab_{dot}(\mathbf{y})$, iff

$$\sum_{\omega_i \models \{a_1, ..., a_n, b\}} \delta^{2\mu(\omega_i)} \geq \sum_{\omega_i \models \{a_1, ..., a_n \neg b\}} \delta^{(1+2\mu(\omega_i))} \tag{9}$$

Let $m^+ = \max\{\mu(\omega_i)|\omega_i \models \{a_1, ..., a_n, b\}\}$ and $m^- = \max\{\mu(\omega_i)|\omega_i \models \{a_1, ..., a_n, \neg b\}\}$, where we define $m^- = -1$ if $\{a_1, ..., a_n, \neg b\}$ is inconsistent (i.e. if $b = \neg a_i$ for some $i$). Then we have $\Theta \models a_1 \wedge ... \wedge a_n \rhd b$ iff $m^+ > m^-$. If $m^+ > m^-$, we find

$$\sum_{\omega_i \models \{a_1, ..., a_n, b\}} \delta^{2\mu(\omega_i)} \geq \delta^{2m^+} = \delta \cdot \delta^{1+2(m^+-1)} > 2^{|\mathcal{A}|} \cdot \delta^{1+2(m^+-1)} \geq 2^{|\mathcal{A}|} \cdot \delta^{1+2(m^-)}$$

$$\geq \sum_{\omega_i \models \{a_1, ..., a_n, \neg b\}} \delta^{(1+2\mu(\omega_i))}$$

Now, conversely, suppose $m^+ \leq m^-$. Then we have

$$\sum_{\omega_i \models \{a_1, ..., a_n, b\}} \delta^{2\mu(\omega_i)} \leq \sum_{\omega_i \models \{a_1, ..., a_n, b\}} \delta^{2m^+} \leq 2^{|\mathcal{A}|} \cdot \delta^{2m^+} < \delta \cdot \delta^{2m^+} = \delta^{1+2m^+} \leq \delta^{1+2m^-}$$

$$\leq \sum_{\omega_i \models \{a_1, ..., a_n, \neg b\}} \delta^{(1+2\mu(\omega_i))}$$

where the last step relies on the fact that $m^+ \leq m^-$ implies $m^- \geq 0$, and hence $\{a_1, ..., a_n, \neg b\}$ must be consistent. We thus have that $m^+ > m^-$ is equivalent to $\sum_{\omega_i \models \{a_1, ..., a_n, b\}} \delta^{2\mu(\omega_i)} \geq \sum_{\omega_i \models \{a_1, ..., a_n \neg b\}} \delta^{(1+2\mu(\omega_i))}$, which is equivalent to $b \in Lab_{dot}(Emb_{had}(a_1, ..., a_n))$.

