# OpenReview forum: "Modelling Monotonic and Non-Monotonic Attribute Dependencies with Embeddings: A Theoretical Analysis"
_AKBC.ws/2021/Conference — AKBC 2021_

### Official Review · Reviewer_wDBJ · 2021-07-21
**Decent paper with a good contribution. However, the writing can be significantly improved.**

**Rating:** 6
**Confidence:** 1

**Review:**

This paper looks at entity embeddings and theoretically tries to answer which kinds of semantic dependencies can be modeled by them. Mainly focusing on settings where the entity embeddings are obtained by pooling the embeddings of its attributes (although this is a strong assumption), this paper presents various theoretical limitations of the embedding approaches in learning logical/structured attribute representations with known logical or structured dependencies in practice. The paper shows that some of the most popular embedding models are not able to capture basic logical rules.

I am not an expert in this topic. However, some of the assumptions made in the paper seem too strong to me e.g. that entity embeddings are obtained by pooling the embeddings of its attributes seems like a strong assumption. Most popular embedding approaches do not seem to follow this assumption. I also found the paper hard to follow. The authors can perhaps work a bit on making this paper accessible to those not working exactly on this problems. For example, the assumptions and the results and how it relate to well known entity embedding approaches can be made more clear. The introduction is too long and already presents technical details, so rewriting it might help.

---

> ### Author Response · Authors · 2021-07-27
> **Response to review**
>
> The assumption that entity embeddings are obtained by pooling attribute vectors indeed restricts the scope of the framework. However, this setting still covers a wide range of popular approaches. Please also see our response to Reviewer 2.
>
> We will attempt to make the paper more accessible, and would welcome any further suggestions on this point.

---

### Official Review · Reviewer_ceQ9 · 2021-07-22
**Excellent paper exploring fundamental questions related to representational capacity**

**Rating:** 9
**Confidence:** 5

**Review:**

### Summary
This paper provides a thorough analysis of the representational capacity of KB embedding models, analyzing the extent to which they can capture logical dependencies of a knowledge base. The authors consider a large range of common scoring functions, and both monotonic and non-monotonic dependencies. The authors make salient points as to the relevance of this sort of analysis - i.e. if a model is incapable of representing such dependencies then it cannot possible capture them from data - however there are a few limitations. For one, the authors are focusing on a setting where entity embeddings are pooled representations of attribute representations, which is not particularly standard (as acknowledged by the authors). Furthermore, there is a question of the extent to which the counterexamples as presented often manifest in real data, and how poorly a model which compromises to meet its objective would perform on various relevant queries. Both of these are reasonable limitations, however, in that they make the theoretical analysis possible and suggest conclusions that may reasonably apply to real-world data settings.

### Questions

After equation (5): "Here, we need to add this term, which amounts to imposing a Gaussian prior, to ensure that $Emb_{sig}$ is well-defined." Is it trivial to see that this actually ensures $Emb_{sig}$ is well-defined?


### Suggestions

For the counterexamples, it may be clearer to read if they are presented as statements and proofs, for example:

> Counterexample #: The knowledge base $K = \{\text{ (rules) }\}$ cannot be represented using $E_\text{something}$, $L_\text{something}$.
> **Proof:** (details here)

The statement that the set of $K$ which are representable using $Emb_{norm}$ are equivalent for $Lab_{dot}$ and $Lab_{dist}$ (following Counterexample 3) could be made into a proposition.

### Typos

After Counterexample 2: "result" -> "results"

---

> ### Author Response · Authors · 2021-07-27
> **Response to review**
>
> Our setting does not cover approaches where entity embeddings are learned from text descriptions using deep neural networks, for instance, but the idea of learning entity embeddings by pooling attribute vectors is nonetheless rather common. First, this setting encompasses skipgram-like models, which remain popular, even in combination with transformer models. For instance, [1] proposes a model for representing entities in which “attribute vectors” correspond to embeddings of sentences that mention the entity. It follows from our analysis of the sigmoid based embeddings that this model cannot capture arbitrary propositional dependencies. Second, averaging based approaches are commonly used in NLP (e.g. representing topics as weighted averages of document vectors, or document/sentences as weighted averages of word vectors). Finally, while we have focussed on the propositional setting, the limitations we identified are also relevant for knowledge graph embedding. This is clearly the case for inductive knowledge graph embedding methods, which often proceed by computing the embedding of an unknown entity by inferring an “attribute vector” from each of the triples mentioning that entity, and then taking a weighted average of these vectors. Moreover, many standard knowledge graph embedding methods essentially proceed by iteratively updating entity vectors in a similar fashion. We therefore suspect that the results from our paper can be used to identify theoretical limitations of standard knowledge graph embedding models, although a formal analysis of such limitations is left for future work.
>
> We agree that the theoretical limitations we have identified may not tend to be problematic in practice, although this will be task-dependent. Our personal motivation for studying these limitations is inspired by applications where embeddings are used as knowledge bases, where guarantees on expressiveness are perhaps more important. Please also see our response to the first review on this point.
>
> The regularisation term in $\textit{Emb}_{\textit{sig}}$ ensures that the maximum is attained, which was the intended purpose. However, this does not guarantees uniqueness, which we will clarify in the final version; if multiple maxima exist, the results hold regardless of which one is chosen.
>
> We appreciate the suggestions for improving the readability of the paper.
>
> [1] Jeffrey Ling, Nicholas FitzGerald, Zifei Shan, Livio Baldini Soares, Thibault Févry, David Weiss, Tom Kwiatkowski:
> Learning Cross-Context Entity Representations from Text. CoRR abs/2001.03765 (2020)

---

### Official Review · Reviewer_mZbd · 2021-07-23
**Interesting Theoretical Analysis on Different Models.**

**Rating:** 7
**Confidence:** 3

**Review:**

**Paper summary**:
This paper tackles the problem of modeling monotonic and non-monotonic reasoning rules in different embeddings spaces, mostly vectors, but under different score functions used by the vector embeddings. The paper first provides the definition of monotonic reasoning, then iterates through a few vector-based embedding and label functions to check if they are able to model the monotonic reasoning part. Then the paper proposed a Relu based model that is able to model both monotonic and non-monotonic reasoning.


**Strength**:
- This paper did a good theoretical analysis on monotonic and non-monotonic kinds of reasoning for word embeddings.
- They iterate different embeddings and label functions for Euclidean vector embeddings. For cases that the proposed methods did not work, they give enough counterexamples and provide proof.
- The paper also includes a Relu based model that is able to model both monotonic and non-monotonic reasoning.

**Weakness / Questions / Suggestions**
- There are many counterexamples in the paper to show that existing embedding and score functions can not model monotonic reasoning. Still, there are limited real-world examples to reflex the effect except for the one example after counterexample 2. If such counterexample does not or seldom exist in real data, the claim and the effect of the counterexample will be weakened.
- The different combinations of the vector embedding model and label score function are limited.
- The writing of the paper is a bit hard to follow, probably due to my limited familiarity with propositional logic. But it would be helpful to refine the writing, especially the part at the end of the introduction where monotonic reasoning and non-monotonic reasoning are first brought up. One specific suggestion is that it would be helpful to move the example of “parent, female and mother” right after the definition of the specific embedding and label functions to make it easier to make the connection of why unwanted dependencies are created due to these functions.
- Echoing the first point, this is a theoretical analysis paper, in which a lot of the examples are hypothetical. Whether such reasoning exists in real data or how much it is there in real data remains a question. In addition, for models where you can model monotonic and non-monotonic reasoning, it remains a question of how easy it is to reach the perfect solution since none experiments are performed, not to mention the Relu function tends to have dead gradient issue which makes training harder.

---

> ### Author Response · Authors · 2021-07-27
> **Response to review**
>
> Regarding the real-world relevance of the counterexamples, one point to note is that Counterexample 3 and its variants rely on a more intricate set of dependencies than Counterexample 2. This suggests that the theoretical limitations of the corresponding strategies might indeed be somewhat less problematic in practice (compared to the strategy underpinning Counterexamples 1 and 2). However, while the types of reasoning that are required for Counterexample 3 may feel somewhat artificial, there could be other sets of dependencies that cannot be modelled with the corresponding strategies either, and which are more likely to occur in practice. More generally, the extent to which the theoretical limitations of particular strategies are problematic in practice is likely to be application-dependent.
>
> Regarding the considered combinations of embedding and labelling functions, clearly the analysis cannot be exhaustive in this respect. However, the considered combinations include the most commonly used approaches, including averaging based strategies and skipgram-like embeddings. Moreover, it should be noted that some of the results could straightforwardly be generalised. For instance, the Sigmoid based embeddings only rely on the fact that entity embeddings are a conical combination of attribute embeddings. The labelling functions include the common case where properties are predicted using a linear classifier (or a sigmoid layer) and distance a based approach.
>
> We appreciate the suggestion for improving the readability of the paper.
>
> It is indeed the case that strategies which can successfully simulate monotonic and non-monotonic reasoning in theory may not work well in practice. The aim of these “positive” results is merely to show that it is possible to simulate reasoning, even for relatively simple embedding strategies, rather than the recommend particular embedding strategies. In fact, it should still be possible to simulate monotonic and non-monotonic reasoning when replacing ReLU by sigmoid, although the proof would be “messier”. Indeed, for the entity embedding $(y_1,…,y_{l+1})$ in the proof of Proposition 1, we have that that $\sigma(y_i)$ can be made arbitrarily close to 0 by choosing $\delta$ sufficiently large. Many other types of non-linearities can be used as well. For instance, if we guarantee that all coordinates are strictly positive, we can choose $f(x_1,…,x_{l+1}) = f(x_1^{-k},…,x_{l+1}^{-k})$ for $k$>1, using again a very similar argument to the one in Proposition 1.

---

### Decision · Program_Chairs · 2021-08-17

**Decision:**

Accept

**Comment:**

This paper presented a theoretical analysis on the representation capacity of entity embeddings based on their attributes ( i.e. modeling the semantic dependencies between attributes), in particular monotonic and non-monotonic attribute dependencies. In general, the reviewers appreciate the theoretical contribution of this paper. We are glad to accept the paper in the AKBC program, though it is strongly recommended for the authors to improve the readability of the paper according to the suggestions by the reviewers.